# Exploring the Limits of Early Predictive Maintenance in Wind Turbines Applying an Anomaly Detection Technique

**DOI:** 10.3390/s23125695

**Published:** 2023-06-18

**Authors:** Mindaugas Jankauskas, Artūras Serackis, Martynas Šapurov, Raimondas Pomarnacki, Algirdas Baskys, Van Khang Hyunh, Toomas Vaimann, Janis Zakis

**Affiliations:** 1Department of Computer Science and Communications Technologies, Vilnius Gediminas Technical University, Saulėtekio al. 11, LT-10223 Vilnius, Lithuania; arturas.serackis@vilniustech.lt (A.S.); martynas.sapurov@ftmc.lt (M.Š.); raimondas.pomarnacki@vilniustech.lt (R.P.); algirdas.baskys@vilniustech.lt (A.B.); 2State Research Institute Center for Physical Sciences and Technology, Sauletekio Av. 3, LT-10257 Vilnius, Lithuania; 3Department of Engineering Sciences, University of Agder, Postboks 422, 4604 Kristiansand, Norway; huynh.khang@uia.no; 4Department of Electrical Power, Engineering and Mechatronics, Tallinn University of Technology, Ehitajate Tee 5, 12616 Tallinn, Estonia; toomas.vaimann@taltech.ee; 5Institute of Industrial Electronics and Electrical Engineering, Riga Technical University, 12/1 Azenes Street, LV-1048 Riga, Latvia; janis.zakis@rtu.lv

**Keywords:** SCADA, wind turbine, anomaly, temperature, neural network

## Abstract

The aim of the presented investigation is to explore the time gap between an anomaly appearance in continuously measured parameters of the device and a failure, related to the end of the remaining resource of the device-critical component. In this investigation, we propose a recurrent neural network to model the time series of the parameters of the healthy device to detect anomalies by comparing the predicted values with the ones actually measured. An experimental investigation was performed on SCADA estimates received from different wind turbines with failures. A recurrent neural network was used to predict the temperature of the gearbox. The comparison of the predicted temperature values and the actual measured ones showed that anomalies in the gearbox temperature could be detected up to 37 days before the failure of the device-critical component. The performed investigation compared different models that can be used for temperature time-series modeling and the influence of selected input features on the performance of temperature anomaly detection.

## 1. Introduction

Predictive maintenance of wind turbines is a critical aspect of wind energy management that involves using data analysis and machine learning techniques to predict when maintenance tasks will be required for wind turbines. It is important because it helps wind farm operators reduce the downtime and the repair costs, increase the operational efficiency of their wind turbines, and ensure the safety of their workforce. With the increasing global demand for clean energy and the growing number of wind turbines in operation, predictive maintenance has become a vital tool for wind energy companies to improve their bottom line and meet their sustainability goals. Furthermore, as predictive maintenance technology continues to advance, wind farm operators can remotely monitor their turbines, reducing the need for manual inspections and improving safety for maintenance workers.

Several components of wind turbines are known to fail more frequently than others, including the following: gearboxes, bearings, blades, and control systems.

Gearboxes are critical components that convert the low-speed rotation of the blades into high-speed rotation for the generator. Due to the high stresses and heavy loads involved, gearboxes are prone to wear, overheating, and oil contamination, which can cause failure. The bearings support the rotor and other rotating parts of the wind turbine. They are subjected to high loads and rotational speeds, making them prone to wear and damage. The generator, as a key component of the wind turbine, converts mechanical energy into electrical power. The high value generator accounts for about 10% of the total cost of the wind turbine [1]. Another significant contributor to wind turbine downtime is the generator faults. Mechanical, electrical, and cooling system failures are the three basic types of generator failures. Typically, all of these types of failure result in the overheating of a generator and the downtime of a wind turbine [2,3].

Wind turbine blades are exposed to extreme environmental conditions, including high winds, temperature changes, and lightning strikes. Over time, this can lead to blade erosion, cracking, and other forms of damage that can affect their performance. The control systems of a wind turbine are responsible for regulating its operation and ensuring that it operates within safe and optimal parameters. If these systems fail or malfunction, the turbine may be shut down or run at suboptimal levels, leading to reduced efficiency and increased maintenance costs.

The application of predictive maintenance techniques is desirable because operators can monitor these and other critical components of their wind turbines to identify potential problems before they occur, allowing them to take proactive steps to address them and avoid costly downtime.

The global installed capacity of wind power has increased from approximately 2.5 GW in 1992 to almost 236 GW at the end of 2021. The plan is to install 116 GW of new wind farms in Europe over the period 2022–2026, which is 23 GW a year on average [4,5]. Since the number of wind farms is growing rapidly, manual monitoring or periodic maintenance of wind turbines is inefficient, which is another reason why automatic monitoring and sensor data analysis techniques are the better option. Mapping wind turbine sensor data and the ambient conditions, such as temperature, wind speed, or humidity, to a failure of the particular wind turbine component introduces several very important challenges. First, we should provide a sufficient amount of examples that carry information about the dependencies of the sensor measurements and the failure event. Second, machine learning models should be designed to deal with highly unbalanced data, as there are hundreds or thousands of measurements with only a few failure events in the data set. Third, there is no prior information on how early we can predict a failure from sensor data, because we cannot directly see which sensor measurements (or extracted features from the sensor signals) start to predict the failure of the component and its timing.

In this paper, we focus our investigation on machine learning techniques that could be efficiently applied for the analysis of wind turbine sensor data and to indicate anomalous sensor measurements related to the possible failure of the wind turbine components. The anomaly detection-based approach in this paper is also used to indicate how many days in advance we may expect to detect the first signs of abnormal conditions in the components of the wind turbine, which will be later followed by failure.

It is important to predict wind turbine failure as early as possible because it allows wind farm operators to plan and schedule maintenance tasks in advance, minimizing the downtime and reducing the risk of costly repairs or equipment replacement. By predicting failures in advance, operators can order replacement parts, allocate resources, and plan maintenance work without disrupting the operation of the wind farm or risking the safety of maintenance workers.

In addition, predicting failures in advance can help wind farm operators optimize their maintenance schedules and reduce maintenance costs. Rather than relying on scheduled maintenance at fixed intervals, predictive maintenance enables operators to perform maintenance only when it is needed, based on the actual condition of the wind turbine components. This can help extend the useful life of the equipment, reduce repair costs, and improve the overall efficiency of the wind farm.

Finally, predicting failures in advance can also help improve safety for maintenance workers. By identifying potential problems before they occur, operators can take steps to minimize the risk of accidents or injuries during maintenance work, ensuring that workers are able to perform their duties safely and efficiently.

The remainder of the paper is organized as follows. In Section 2, we present the state of the art. In Section 3, we present the proposed technique. In Section 4, we discuss the building blocks of the deep anomaly detection framework and we describe the case study and the results obtained; finally, in Section 5, we summarize the present work and draw our conclusions.

## 2. State of the Art

The field of predictive maintenance of wind turbines is rapidly advancing, with new technologies and techniques being developed to improve the accuracy and efficiency of predictive maintenance systems. In this section on the state of the art, we explore some of the latest developments in the field, including advances in machine learning algorithms, the use of sensors and IoT devices, and the application of big data analytics to wind turbine maintenance. We also examine some of the challenges that remain in the field, such as the need for more accurate data collection and analysis and the need for standardized approaches to predictive maintenance across different wind turbine models and manufacturers. Overall, this section provides an overview of the current state of the art in the predictive maintenance of wind turbines and highlights some of the most promising areas for future research and development.

Most wind farms today are equipped with a Supervisory Control and Data Acquisition (SCADA) system, and the SCADA data are full of hidden information that can be utilized to detect faults early on. SCADA signal research has also been successful in establishing systems to identify wind turbine failures [6,7,8,9]. The generator temperature and gearbox oil temperature in SCADA data as the entry point for fault warning were used in [10].

Big data are promising for modeling wind energy, including wind speed forecasting [11,12], power prediction and optimization [12], power curve monitoring [13], and predictive maintenance [14,15]. Appropriate use of condition monitoring (CM) can reduce turbine repair and maintenance costs by detecting faults at an early stage [2]. Several papers have reviewed CM methods for various components of wind turbines [7,16,17]. The most popular models were based on Multilayer Perceptron (MLP), Decision Tree (DT), k-Nearest Neighbor (kNN), Support Vector Machine (SVM), Long Short-Term Memory (LSTM), etc. In addition, early stage faults on the gear teeth were identified during condition monitoring using vibration analysis [18,19].

The authors of [15] stated that the MLP stacked model reduced wind turbine maintenance costs and predicted failures with 80.53% precision. This best result was achieved compared to the base models. Stacked algorithms rarely produced unanimously better results than the best of the two base models in terms of the metrics used. However, stacking multiple neural networks had this effect. The final stacked model produced more true positives and true negatives, while at the same time, it reduced the number of false negatives.

A data-driven decision-making methodology that combined prognostic and health management, proposed by [20], reduced maintenance costs, as compared to the stacked MLP model from [15] with a precision of 91.08%. Data processing, feature engineering, and model development were performed before model selection and validation. The best result was achieved when a specific model for each failure was chosen. In this model, the 2016 data for all turbines were combined into one training dataset, and then a training set was created for each failure of each component by assembling the failure data related to this component of all turbines. The data balance chosen here was 25% failure and 75% without failure, with the 75% randomly selected from all turbines. It should be noted that the support vector classifier algorithm was dropped due to its long computation time, and the random forest algorithm was chosen for the testing phase due to the training and validation accuracy percentages.

A data-centric ML technology was proposed, where data-oriented steps were not a one-time procedure but were executed iteratively [14]. It is important to note that the ML model was not time dependent, and the data were randomly sampled from the remaining half of the data from the testing period and used as a validation set.

The detection of anomalies in time series is a highly complex task that requires large data sets to accurately discriminate between points that exhibit normal and abnormal behavior [21]. For example, the authors of [22] tried to determine the icing of wind turbine blades. Their method of choice was the use of LSTM networks because of their capability to process and analyze sequential data effectively. Using these LSTM networks, the researchers classified the condition of the turbine blades into two scenarios: blades with ice and blades without ice. The LSTM model was trained to interpret sequential patterns using a combination of historical data, real-time measurements, and possibly other operational variables such as weather conditions and performance data. The authors reported a remarkable precision of 96% in detecting icing anomalies using an LSTM network.

The authors of [23] claimed that they were able to detect four types of anomalies (generator failure, pitch motor failure, gearbox oil temperature overlimit, and anemometer failure) with 91.29% precision, considering 25 different parameters. The authors of [23,24,25,26,27,28,29] used SCADA data for anomaly detection. In these cases, data sets of 10 to 14 measured metrics were used. In all the works, good results were provided for the detection of anomalies in wind turbines; for example, in [30], the temperature of the gearbox was monitored, and the authors were able to detect possible faults before they occur. Furthermore, ref. [29] used a semi-supervised model using only healthy information to develop this strategy, and they claimed that it was not necessary to have a history of turbine failure. It can be concluded that the above methodology worked well and efficiently, as it raised alerts months before fatal breakdowns when tested on total wind parks over a year without false alarms.

The experiments described in [10] showed that in terms of XGBoost compared to the SVM regression prediction mode, the fixed fault threshold approach could provide a fault alarm 3 h in advance for the generator, while the dynamic fault threshold determination method could provide a fault alarm 4.25 h in advance. The fixed fault threshold approach could provide a fault alarm 2 h in advance for a gearbox, but the dynamic threshold fault diagnosis method may provide a fault alarm 2.75 h in advance.

The authors of [26] claimed that the Mahalanobis distance approach detected faulty behaviors of wind turbines at a high accuracy rate of 97%. For this study, a direct-drive onshore wind turbine located in Taiwan was used. The duration points for anomaly detection were averaged for the first 12 h of the day and for the following 12 h. Regarding the parameters of different components, the fact that the MD approach succeeded in detecting wind turbine faults suggests that it can provide a comprehensive view of the system’s health.

The original unsupervised deep anomaly detection framework with the core of a neural architecture combining AEs and Graph Convolutional Networks (GCNs) was proposed by [31]. The results showed that the proposed model could anticipate 10 SCADA log alarms with an average failure time of about 23 days involving some of the most critical components, without triggering false alarms.

The majority of these algorithms were built in an artificial environment and were often not highly robust to the failure behavior of real-world systems, i.e., minor system failures (e.g., a failing sensor, small damage, or firmware updates). In most cases, even a small fault causes the engine to fail completely, resulting in the continuous reporting of an anomaly or the inability to detect additional anomalies [32].

## 3. Materials and Methods

### 3.1. The Dataset

We used a public Energias De Portugal (EDP) wind turbine dataset from the Wind Turbine Failure Detection Challenge [1]. This data set covered two full years of SCADA records from 2016 to 2017 from five wind turbines at 10-minute intervals and data from the meteorological mast. The data set was divided into two data sets: 80% training data and 20% testing data. Furthermore, this data set contained the failure list of the training and the testing sets, as presented in Table 1. Those failures were grouped by component: gearbox, generator, generator bearing, transformer, and hydraulic group. In this data set, only a few turbine components had failures in both the training and testing data sets. The wind turbine T01 had only one gearbox failure in the training data and only one in the transformer. The data from this turbine were good for training our model. The T06 wind turbine had many generator failures in the training set but only one failure for the gearbox group and one for the hydraulic group. The wind turbine T07 had two transformer failures and one generator bearing failure in the training set and two hydraulic group failures, one generator failure, and one generator bearing failure. The wind turbine T09 had three gearbox bearings failures and one generator failure in the training set and one each in the gearbox, generator bearing, and hydraulic groups. The T11 wind turbine had one failure in the gearbox and one in the hydraulic group in the training set and two failures in the hydraulic group.

Unbalanced data can cause many failures. In this case, we decided to monitor the most expensive part: the gearbox. In the following subsections, the different predictive maintenance approaches with this data set are reviewed.

### 3.2. The Proposed Technique

In this paper, we propose an approach that is less sensitive to the data-imbalance problem. We followed the hypothesis that the temperature of the gearbox bearing on the high-speed shaft correlated with all five failure types in the dataset and could be predicted in advance if we noticed unusual changes in the measured temperature.

One challenge in temperature anomaly detection is dynamic temperature changes, which depend on the load of the gearbox and the current ambient temperature. Therefore, we could not use a fixed temperature level as an indicator of the anomaly. We decided to use a machine learning model for temperature prediction, which should continuously predict the temperature of the gearbox, and we compared it to the measured temperature. The model should be able to predict the temperature with a minimum error when all the turbine components are in good condition. This was achieved by training a machine learning model to map the measured sensor data to the gearbox temperature. Such a model always indicates what temperature we should expect in the current situation (with the current ambient temperature, wind speed, etc.). If the model works correctly, and the real measured temperature is significantly different from the predicted one, then an anomaly of the temperature values is indicated, and we may predict the failure of the wind turbine component.

According to the experience and knowledge of experts, we filtered out a set of ambient (e.g., wind speed and temperature) and internal signals (e.g., generator RPM) to predict the gearbox bearing at a high-speed shaft temperature in a normal state when there were no failures in wind turbines. All these signals are listed in Table 2.

The implementation and practical application of our approach, proposed in this paper, needed to perform the following investigations:Selection of the input features for a temperature prediction model that sufficiently represented the influence of external factors on the gearbox bearing temperature when the turbine components were in a healthy state.Selection of the machine learning model to solve the temperature prediction task with a minimized regression error.Proposal of a technique for anomaly detection threshold selection when the predicted temperatures were compared with the actual measured ones.

These features were sampled in the range of 3, 6, and 24 h. The selection of these particular time frames was guided by domain experience and the findings from previous research in the field of the predictive maintenance of wind turbines.

Recurrent Neural Networks (RNNs) are a class of neural networks that are particularly suited for time-series data analysis. RNNs are designed to handle sequences of data with varying lengths and can capture temporal dependencies. This is essential when working with time-series data, such as wind turbine sensor data, as it allows the model to learn patterns that span across different time steps. In the study, the gearbox bearing temperature was predicted with three different models: Long-Short-Term Memory (LSTM), Gated Recurrent Unit (GRU), and Bidirectional Long Short-Term Memory (bi-LSTM). These results are shown in Figure 1.

Structures with a recurrent layer (128 cells) and two recurrent layers (128 + 100 cells) were tested.

Training and testing were applied multiple times with different scenarios when the training data were selected from one turbine and tested with data from other turbines. In addition, multiple turbine data sets were combined and tested with the remaining data. The regression models were trained and tested 144 times to select the model with the best RMSE score.

## 4. Results and Discussion

This study offers substantial contributions to the field of predictive maintenance and failure detection, particularly in the context of wind turbine operations. The successful implementation of a recurrent neural network, specifically a bidirectional long short-term memory (bi-LSTM) model, highlights the model’s ability to effectively predict temperature anomalies, serving as a potential indicator of imminent gearbox failure. This predictive capability allows for the timely mitigation of equipment failure, enhancing the efficiency and productivity while minimizing the downtime.

The comparative analysis also revealed important insights. The superior performance of the two-layer bi-LSTM model over the other regression models reaffirmed the efficacy of recurrent neural networks in time-series prediction tasks. The models’ performances, however, were significantly influenced by the choice of input features. Shorter time ranges for the input features provided better predictive accuracy, underlining the importance of data granularity in time-series prediction. It achieved the lowest Root Mean Square Error (RMSE) at 1.61, indicating a strong predictive precision. This model also demonstrated the lowest standard deviation (Std) at 1.10, suggesting consistent performance. Furthermore, it had the smallest minimum error (Min) at 0.0021 and the smallest maximum error (Max) at 7.17, demonstrating its robustness in handling a variety of prediction scenarios. The results are displayed in Table 3

The models were trained to predict the gearbox bearing at a high-speed shaft temperature in order to compare it with the actual temperature to find anomalies. Two automatic THD selection techniques were compared, based on three standard deviations from the mean value and based on the moving median. The effect on the early failure prediction is given in Table 4. The selection of the decision threshold is illustrated in Figure 2.

However, while our findings are promising, they also raise additional questions for future research. For example, how would the bi-LSTM model perform with different parameter configurations or in predicting other types of anomalies? Furthermore, while shorter time intervals proved beneficial in this context, further research is needed to establish the optimal time interval that balances the prediction accuracy and computational efficiency.

## 5. Conclusions

In this investigation, we demonstrated the potential of recurrent neural networks, particularly a bidirectional long short-term memory (bi-LSTM) model, in effectively predicting temperature anomalies that may signify imminent failures in wind turbine gearboxes. Our model’s ability to accurately model the time series of a healthy device’s parameters has proven instrumental in detecting these anomalies by comparing the predicted values with actual measurements.

The comparative analysis between various models and input features further established the superiority of a two-layer bi-LSTM model and the advantages of using shorter time ranges for input features (3 h). This outcome illuminates the importance of selecting an appropriate model and the right time granularity in time-series prediction tasks.

Our approach using THD selection based on the moving median has shown promising results in facilitating the detection of anomalies 60% earlier than other techniques. This capacity for early detection underscores the potential of our method as an effective predictive maintenance tool, enabling the proactive mitigation of equipment failures and thus conserving time and resources.

## Figures and Tables

**Figure 1 sensors-23-05695-f001:**
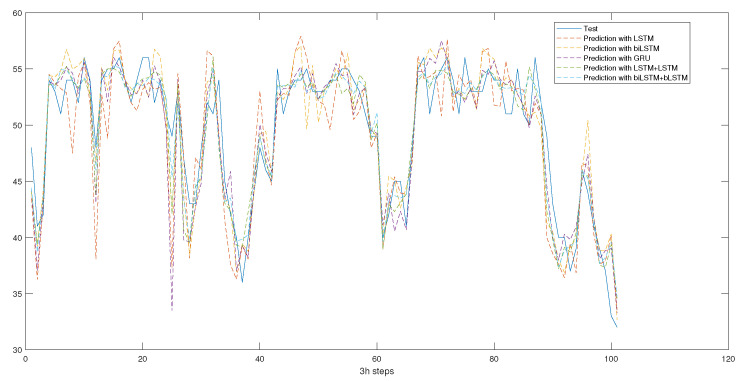
T01, T03, and T09 actual vs. predicted measurements with different models.

**Figure 2 sensors-23-05695-f002:**
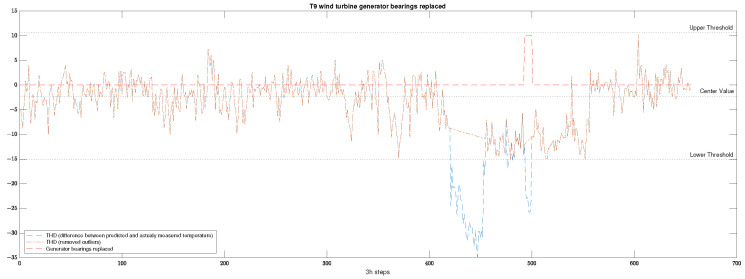
Illustration of a selected decision threshold for temperature anomaly detection.

**Table 1 sensors-23-05695-t001:** Failure list based on [1].

Component	T01		T06		T07		T09		T11	
	Train	Test	Train	Test	Train	Test	Train	Test	Train	Test
Gearbox	1	0	0	1	0	0	1	1	0	0
Generator	0	0	5	0	0	1	0	0	1	0
Generator Bearing	0	0	0	0	1	1	3	1	0	0
Transformer	0	1	0	0	2	0	0	0	0	0
Hydraulic Group	0	0	1	1	0	2	0	1	1	2

**Table 2 sensors-23-05695-t002:** List of the ambient and internal signals used for predicting the gearbox bearing temperature of wind turbines.

Descriptor	Description	Component
Gen_RPM_Max [rpm]	Maximum generator rpm in latest average period	Generator
Gen_RPM_Min [rpm]	Minimum generator rpm in latest average period	Generator
Gen_RPM_Avg [rpm]	Average generator rpm in latest average period	Generator
Gen_RPM_Std [rpm]	Std generator rpm in latest average period	Generator
Rtr_RPM_Max [rpm]	Maximum rotor rpm in latest average period	Rotor
Rtr_RPM_Min [rpm]	Minimum rotor rpm in latest average period	Rotor
Rtr_RPM_Avg [rpm]	Average rotor rpm in latest average period	Rotor
Amb_WindSpeed_Max [m/s]	Maximum wind speed within average timebase	Ambient
Amb_WindSpeed_Min [m/s]	Minimum wind speed within average timebase	Ambient
Amb_WindSpeed_Avg [m/s]	Average wind speed within average timebase	Ambient
Amb_WindSpeed_Std [m/s]	Std wind speed within average timebase	Ambient
Amb_WindDir_Relative_Avg [${}^{\circ }$]	Average wind relative direction	Ambient
Amb_WindDir_Abs_Avg [${}^{\circ }$]	Average wind absolute direction	Ambient
Amb_Temp_Avg [${}^{\circ }$C]	Average ambient temperature	Ambient
Prod_LatestAvg_ActPweGen0 [Wh]	Active power: generator disconnected (yaw motor, hydraulic motor, etc.)	Production
Prod_LatestAvg_TotActPwr [Wh]	Total active power	Production
Prod_LatestAvg_ReactPwrGen0 [VArh]	Reactive power: generator disconnected (yaw motor, hydraulic motor, etc.)	Production
Prod_LatestAvg_TotReactPwe [VArh]	Total reactive power	Production

**Table 3 sensors-23-05695-t003:** Best model performance metrics.

Model	RMSE	Std	Min	Max	Training Time
LSTM	1.91	1.49	0.0012	11.58	09:22
bi-LSTM	1.79	1.34	0.0033	10.27	06:47
GRU	1.85	1.33	0.0058	10.42	05:41
LSTM + LSTM	1.69	1.09	0.0027	7.04	08:06
bi-LSTM + bi-LSTM	**1.61**	**1.10**	**0.0021**	**7.17**	**11.10**

**Table 4 sensors-23-05695-t004:** Estimation of how many days in advance the failure could be predicted from the anomalies.

Component Failure	Model with Two		Model with Two	
	LSTM Layers		bi-LSTM Layers	
	THD Estimation Method		THD Estimation Method	
	Mean + 3*STD	Moving Median	Mean + 3*STD	Moving Median
Gearbox	14	22	22	32
Generator	15	24	26	37
Generator Bearing	13	24	22	36
Transformer	23	22	24	35
Hydraulic Group	12	22	20	33

* THD selection techniques compared: based on 3 standard deviations from mean value and based on moving median.

## Data Availability

Not applicable.

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
