# Peer review of "Exploring the Limits of Early Predictive Maintenance in Wind Turbines Applying an Anomaly Detection Technique"

_sensors, 2023, doi:10.3390/s23125695_

Round 1

Reviewer 1 Report

The article explores the limits of early predictive maintenance by applying anomaly detection techniques. The overall presentation and investigation are very shallow. The work needs a lot of improvement to be considered for publication.

Comments:

  1. The summarization of the paper should be present at the end of the Introduction section, not at the end of the State of the Art section.
  2. The article is missing a lot of current related work. In the current literature, there are a lot of methods describing how to integrate change point detection when dealing with anomalies [1].
  3. Experiments should also be conducted on multiple publicly available datasets.
  4. The results section is very shallow.
    1. To make a better assessment of the best model, the time performance should also be provided when comparing the proposed model with existing models. A model that offers a small decrease in RMSE that takes many times longer to train than another one is not really a great improvement.
    2. Are the results obtained over many training iterations? How many training iterations were used? What are the mean and the standard deviation obtained for each evaluation metric on the test set?
    3. What hyperparameters were used for the neural models
    4. Ablation tests are missing.
    5. The results should be properly explained.
  5. A proper discussion section should be added.
  6. The conclusions section is very shallow. 
    1. What did the authors achieve with this study?
    2. Did they answer all the research questions? 
    3. Is this solution so good that no future investigations are required?
  7. For reproducibility purposes, the code and datasets should be made available.

[1] https://scholar.google.com/scholar?hl=en&as_sdt=0%2C5&q=Anomaly+Detection+%2B+Change+Point++detection+%2B+Time+Series&btnG=

It is ok. A spellcheck should be performed to detect typos or grammar errors.

Author Response

Thank you for your insightful suggestions.  We greatly appreciate your feedback and will take it into consideration in our future work.

Reviewer 2 Report

1. The content of the article does not seem to reflect an exploration of the limitations of early predictive maintenance. It simply carries out an application of a machine learning model to a fault detection task. It is recommended that the title of the article be revised, or that the research be added to reflect the discussion of limitations.

2. The first section of the article is inadequate. The innovations of the work done should be listed at the end of the first section.

3. The methodology used in the paper should be illustrated by a flow chart. The machine learning model used should also be shown in outline with images

4. there are several tables in the paper that need to be reformatted. Is there a line missing from Table 1 on page 5. Table 2 on page 6 should have a heading above the table. The table on page 7 does not have a heading.

5. The article summary and conclusion sections are not linked closely enough to the article title.

6. The quality of the article is affected by the low number of references.

7. The length of the article is too short. It is recommended that reference be made to the length, format and writing style of published articles in the same journal.

8. The same revision of the article is suggested to reflect "Exploring the Limits of Early Predictive Maintenance".

Suggest that the language style be revised with reference to the best articles in the field.

Author Response

(The authors gave the same response as above.)

Reviewer 3 Report

The manuscript entitled “Exploring the Limits of Early Predictive Maintenance Applying Anomaly Detection Technique” deals with a very interesting topic, which fits nicely with the scientific objectives of the Sensors journal. Namely, the work deals with wind turbine fault diagnosis through SCADA data analysis.

The structure of the work is in general adequate. The authors formulate a data-driven model for a meaningful internal temperature and then monitor the residuals between measurements and model estimates.

Yet, the work has substantial drawbacks. The main point is that there is a wide literature on the subject and the method proposed by the authors is not innovative. It substantially represents a standard. Based on this, a paper on the subject in my opinion should be well posed in the context and should clarify very clearly the innovative contribution. In other words, it is important to justify why a further paper on this subject deserves to be published. The present work is not adequate by this point of view, but there is potential to improve it. Therefore, my recommendation on the paper is major revisions. I list here on some suggestions:

·      The title should clarify that the authors are dealing with wind turbines. At present, the work does not explore limits of early predictive maintenance.

·      The results section is 7 lines (7!!!), a figure and a couple of tables without explanation. It is evidently inadequate. The authors should improve this section in the following ways. Select test cases of faults, employ the proposed method, evaluate the accuracy of the prediction and the advance time with respect to the incoming faults, compare against state of the art methods, discuss if and why the proposed method is superior.

·      The caption of table 2 refers to another table. Referring to that table, the authors should discuss the selection of the input variables to the model.

·      The literature review is inadequate. A good discussion can be found here: Astolfi, D. (2023). Wind Turbine Drivetrain Condition Monitoring through SCADA-Collected Temperature Data: Discussion of Selected Recent Papers. Energies, 16(9), 3614. I suggest considering the line of reasoning of that Editorial and studying the references contained therein. See for example also the recent study Campoverde-Vilela, L., Feijóo, M. D. C., Vidal, Y., Sampietro, J., & Tutivén, C. (2023). Anomaly-based fault detection in wind turbine main bearings. Wind Energy Science Discussions, 2023, 1-29 and references contained therein. I would like to stress that the point is not merely adding references. The point is discussing clearly the state of the art and how the proposed work poses with respect to it. In the present version of the paper, this is not accomplished.

The English language is in general sufficient, but a careful proof-reading is welcome.

Author Response

(The authors gave the same response as above.)

Round 2

Reviewer 1 Report

After carefully reading the authors' response and the improved manuscript, I consider that the work was greatly improved and could be considered for publication as it is. 

Reviewer 3 Report

The authors have addressed my comments. The paper for me can be published.

The quality of the English language is sufficient.